# Deletion Mutants of *Francisella* Phagosomal Transporters FptA and FptF Are Highly Attenuated for Virulence and Are Protective Against Lethal Intranasal *Francisella* LVS Challenge in a Murine Model of Respiratory Tularemia

**DOI:** 10.3390/pathogens10070799

**Published:** 2021-06-24

**Authors:** Brandi E. Hobbs, Courtney A. Matson, Vasileios I. Theofilou, Tonya J. Webb, Rania H. Younis, Eileen M. Barry

**Affiliations:** 1Center for Vaccine Development and Global Health, University of Maryland School of Medicine, Baltimore, MD 21201, USA; bhobbs@som.umaryland.edu; 2Department of Microbiology and Immunology, University of Maryland School of Medicine, Baltimore, MD 21201, USA; courtney.matson@som.umaryland.edu (C.A.M.); twebb@som.umaryland.edu (T.J.W.); 3Department of Oncology and Diagnostic Sciences, School of Dentistry, University of Maryland Baltimore, Baltimore, MD 21201, USA; vtheofilou@umaryland.edu (V.I.T.); ryounis1@umaryland.edu (R.H.Y.); 4The Marlene and Stewart Greenebaum Cancer Center, University of Maryland School of Medicine, Baltimore, MD 21201, USA

**Keywords:** *Francisella tularensis*, Live Vaccine Strain (LVS), live attenuated vaccine, MFS transporter, attenuation, organ burdens, histopathology, cytokines, flow cytometry, C57BL/6J mice

## Abstract

*Francisella tularensis* (*Ft*) is a Gram-negative, facultative intracellular bacterium that is a Tier 1 Select Agent of concern for biodefense for which there is no licensed vaccine. A subfamily of 9 *Francisella* phagosomal transporter (*fpt*) genes belonging to the Major Facilitator Superfamily of transporters was identified as critical to pathogenesis and potential targets for attenuation and vaccine development. We evaluated the attenuation and protective capacity of LVS derivatives with deletions of the *fptA* and *fptF* genes in the C57BL/6J mouse model of respiratory tularemia. LVSΔ*fptA* and LVSΔ*fptF* were highly attenuated with LD_50_ values of >20 times that of LVS when administered intranasally and conferred 100% protection against lethal challenge. Immune responses to the *fpt* mutant strains in mouse lungs on day 6 post-infection were substantially modified compared to LVS and were associated with reduced organ burdens and reduced pathology. The immune responses to LVSΔ*fptA* and LVSΔf*ptF* were characterized by decreased levels of IL-10 and IL-1β in the BALF versus LVS, and increased numbers of B cells, αβ and γδ T cells, NK cells, and DCs versus LVS. These results support a fundamental requirement for FptA and FptF in the pathogenesis of *Ft* and the modulation of the host immune response.

## 1. Introduction

*Francisella tularensis* (*Ft*) is a Gram-negative, facultative intracellular, coccobacillus responsible for the zoonosis tularemia [1]. *Ft* has a broad host range and persists in the environment, causing an average of ~250 natural cases of human tularemia in the U.S. each year [2]. While *Ft* can be contracted by various routes, most commonly through the bite of an insect (ticks) and contact with infected animal products or carcasses, it can be easily aerosolized, and inhalation of as few as 10 colony forming units of virulent *Ft* can cause disease with up to a 60% mortality rate if untreated [3,4,5,6,7,8,9]. *Ft* has a history of weaponization, and combined with the small inoculum, high mortality rate and ease of dissemination, has been listed by the Centers for Disease Control and Prevention as a Tier 1 Select Agent of concern for biodefense. Importantly, there is no licensed vaccine which could protect the population against either naturally occurring tularemia or tularemia resulting from a bioterror attack.

Two subspecies of *Ft*, *Ft* subspecies *tularensis* (Type A) and *Ft* subspecies *holarctica* (Type B), cause disease in humans [10]. Type A *Ft* is the most virulent, and thus, the ultimate vaccine target. Historically, many approaches to vaccine development against tularemia have been utilized, including killed whole cell formulations, subunit vaccines, and live attenuated strains [11,12,13,14,15,16]. To date, the live attenuated approach has been the most extensively studied. The “Live Vaccine Strain” (LVS) is a live attenuated derivative of Type B *Ft* that was first attenuated by former U.S.S.R. scientists and then further attenuated and developed by the U.S. Department of Defense [8,9,17,18]. LVS was tested extensively in human volunteers where vaccination with LVS was shown to provide at least partial protection against aerosol challenge with virulent Type A *Ft*, but protection was dependent upon the route and dose of vaccination and the route and dose of challenge [6,7,19,20,21]. LVS was given investigational new drug (IND) status and was used to reduce the incidence of natural and laboratory-acquired tularemia, yet several shortcomings prevented its licensure by the Food and Drug Administration (FDA) [22,23].

Recently, a new lot of LVS has been produced under current Good Manufacturing Practices (cGMP) and has undergone testing in rabbits and humans where it was found to be safe, well-tolerated, and immunogenic [13,22,24,25]. While LVS remains unlicensed for general use, it provides proof-of-principle that a live attenuated vaccine approach can confer protection against virulent *Ft* [26]. In studies to identify mutations in the genome of LVS that are likely to be responsible for its attenuation, pseudogene-causing mutations were identified in 11 genes, including genes involved in metabolic pathways [27]. Bacteria that are deficient in metabolism or nutrient acquisition often make good live attenuated vaccine candidates because they maintain their surface antigens and the ability to colonize an appropriate niche but are unable to infect efficiently due to a reduction in growth rate [27,28,29]. We similarly hypothesized that a live attenuated strain of *Ft* that expresses critical antigens in their native conformations but contains targeted mutations in genes with well understood roles could result in the generation of a highly attenuated, immunogenic, safe, and efficacious vaccine.

LVS serves as an excellent model for the study of *Ft* pathogenesis in mice. Murine infection with LVS recapitulates virulent human tularemia, including high mortality rates, and the use of LVS allows accelerated study under BSL-2 conditions due to its attenuated virulence in humans [14,30,31]. Furthermore, LVS has 99.2% sequence identity to the virulent Type A SchuS4 strain [27] making it a useful tool to identify targeted mutations that could be generated in the Type A background. Herein we utilize LVS as a model to identify and characterize attenuated *fpt* mutants under BSL-2 conditions before advancing promising candidates in the virulent Type A *Ft* strain under BSL-3 conditions.

Work with *Legionella pneumophila*, a distantly related intracellular bacterium, has implicated the Major Facilitator Superfamily (MFS) of transporters as critical to intracellular pathogenesis and, therefore, potential targets for attenuating mutations [32]. The MFS is the largest transporter family known and is responsible for transporting biomolecules, including sugars and amino acids, and nutrient scavenging [33]. Deletion of the MFS subfamily phagosomal threonine transporter (Pht), PhtA, from *Legionella* rendered the bacterium unable to escape the phagosome and attenuated for virulence [32]. Bioinformatics screens identified 33 MFS-like homologs in *Ft*, and of these, nine belong to an analogous Pht subfamily named the *Francisella* phagosomal transporter (Fpt) subfamily [34]. Previously, we demonstrated that LVS mutants with deletions in three of these genes, *fptB*-an isoleucine transporter [35], *fptE*-an asparagine transporter [36], and *fptG* (substrate unknown), exhibited reduced intracellular replication within macrophages and hepatocytes, and were attenuated for virulence in BALB/c mice [34]. Furthermore, immunization of mice with these mutants conferred protection against lethal challenge with the parental LVS strain [34]. One *fpt* mutation was introduced into the Type A SchuS4 strain to generate SchuS4Δ*fptB*, which displayed altered intracellular replication kinetics in macrophages and was highly attenuated in the C57BL/6J mouse model of tularemia, but disappointingly only provided partial protection against virulent Type A *Ft* challenge [37]. Because this candidate did not provide a high level of protection, our current study focuses on the promising but previously uncharacterized FptA and FptF transporters, which may prove to be better targets for attenuating mutations. In addition to furthering the advancement of an efficacious vaccine against *Ft*, we utilized the LVSΔ*fptA* and LVSΔ*fptF* mutant strains to understand the contributions of these Fpt transporters to the pathogenesis of *Ft*. We found that both *fpt* genes are critical for the induction of a pathological inflammatory response that facilitates pathogenesis and results in host death, and deletion of either gene results in vaccine candidates that provide protection against lethal challenge.

## 2. Results

### 2.1. fpt Mutants Are Attenuated in the C57BL/6J Murine Model of Respiratory Tularemia Compared to Parental LVS

LVS strains with a complete deletion of the *fptA* or *fptF* gene products were constructed utilizing suicide plasmids as previously published [34] to generate the LVSΔ*fptA* and LVSΔ*fptF* mutant strains studied in this work. Previous in vitro studies demonstrated that LVSΔ*fptA* and LVSΔ*fptF* were not deficient for extracellular growth in broth culture or for intracellular replication in J774.1 macrophages, a BALB/c murine macrophage cell line [34]. However, both strains were attenuated for virulence following intraperitoneal (i.p.) injection in BALB/c mice; 100% of animals (n = 4 per group) survived infection with 3 × 10^3^ or 6 × 10^3^ CFU of LVSΔ*fptA* or LVSΔ*fptF*, respectively, compared to no survivors following LVS infection with ~450 CFU (Appendix A). In this current work, we have moved to the C57BL/6J murine model because this mouse strain has been demonstrated to be the most stringent model for protection against lethal *Ft* and is prevalent in the literature for evaluating immune responses to *Ft* [14,38]. Additionally, we have utilized intranasal (i.n.) administration because this more closely mimics respiratory tularemia. In this model, we reassessed the levels of attenuation of the FptA- and FptF-based mutant strains. Groups of C57BL/6J mice were inoculated i.n. with similar doses of parental LVS, LVSΔ*fptA*, or LVSΔ*fptF*, or mock-infected with PBS, and survival was monitored for 28 days or until mice met criteria for euthanasia. An i.n. dose of 350 colony forming units (CFU) of LVS resulted in death of 100% of mice by day 10 post-infection (Figure 1A). In contrast, 100% or 91% of mice survived infection with similar doses of LVSΔ*fptA* or LVSΔ*fptF*, respectively (Figure 1A). 100% of mock-infected mice survived with no associated weight loss or other clinical signs of disease (Figure 1A,B). Whereas LVS-infected mice exhibited lethargy and ruffling of fur and lost > 20% of their initial starting weight (which met euthanasia criteria) by day 10 post-infection with this dose, the LVSΔ*fpt* mutant-infected mice inoculated with a similar dose only lost an average of ~10% of their initial starting weight, exhibited little to no clinical signs, and recovered to mock-infected levels (Figure 1B). Extended dose studies were performed to determine LD_50_ values for each strain. Whereas the LD_50_ value of LVS is <60 CFU, the LVSΔ*fptA* and LVSΔ*fptF* mutant strains were highly attenuated with LD_50_ values of ~1400 CFU, and between ~2010 and 5167 CFU, respectively (Table 1). These results demonstrate that the LVSΔ*fptA* and LVSΔ*fptF* mutant strains are attenuated for virulence in this model compared to LVS, and highlights their potential as vaccine candidates.

### 2.2. Inoculation with LVSΔfptA or LVSΔfptF Confers Protection against Lethal Challenge with Parental LVS in the C57BL/6J Murine Model of Respiratory Tularemia

Given the high level of attenuation of the LVSΔ*fptA* and LVSΔ*fptF* mutant strains (Table 1), we hypothesized that these mutant strains would confer protection against lethal challenge with parental LVS. Surviving mice from the LD_50_ experiments were challenged on day 29 post-infection with a lethal dose of LVS, ~500 CFU i.n., to determine vaccine efficacy. At this challenge dose, there was a 95.8% mortality rate for the unimmunized control group. Control mice exhibited clinical signs of disease, including lethargy and ruffling of fur, and lost > 20% of their initial starting weight, requiring euthanasia, by day 10 post-challenge. In contrast, all immunized mice, regardless of the vaccinating strain and the dose, were protected against lethal challenge with LVS with no clinical signs of disease and little to no loss of weight. A subset of mouse lungs harvested at 21 days post-challenge was tested for challenge strain clearance. No challenge organisms were cultured from the lungs which supports sterilizing immunity conferred by these vaccine strains (Table 1).

### 2.3. fpt Mutants Have Reduced Bacterial Organ Burdens in the Lungs, Livers, and Spleens of C57BL/6J Mice versus Mice Infected with Parental LVS

Next, the mechanisms of attenuation of LVSΔ*fptA* and LVSΔ*fptF* were investigated in the mouse model by first evaluating their ability to colonize the host lung and disseminate to systemic organs, which are characteristics of tularemia that we hypothesized may be compromised in these mutant strains versus LVS. Groups of C57BL/6J mice were inoculated i.n. with ~350 CFU of parental LVS, LVSΔ*fptA* or LVSΔ*fptF*, and organ burdens were enumerated from the lungs (primary site of infection), livers, and spleens on days 1, 3, 6, 14 and 21 post-infection. A dose of ~350 CFU of LVS is 100% lethal but is a well-tolerated dose of the *fpt* mutant strains; this target dose was maintained for all remaining experiments. One day post-infection, the bacterial burdens in the lungs were similar regardless of infecting strain (Figure 2A) which suggests that FptA and FptF are not essential for initial colonization. By day 3 post-infection, the bacterial burdens in the lungs increased significantly as all strains have begun replicating (Figure 2A), and bacteria were present in the spleen (Figure 2B) and liver (Figure 2C) demonstrating that these strains are capable of dissemination to systemic organs. However, by day 6 post-infection, the organ burdens of both mutant strains in the lungs, livers, and spleens were markedly reduced compared to LVS which continued to replicate to high numbers between days 3 and 6 post-infection (Figure 2A–C). Mice that were infected with LVS did not survive past day 10 post-infection, and therefore organ burden data only exists for the *fpt* mutant strains from days 14 and 21 post-infection. Whereas the organ burdens of LVS-infected mice continued to rise through day 6 post-infection, after which time these mice meet euthanasia criteria, the organ burdens of *fpt* mutant-infected mice reached a peak organ burden at day 3 and then steadily declined through 21 days post-infection when most of the bacteria were cleared from mouse organs (Figure 2A–C). These results indicate that FptA and FptF are required for the full virulence of *Ft* and dictate the outcome of a critical host-pathogen interaction that occurs between days 3 and 6 post-infection.

### 2.4. Pathology Is Less Severe in the Lungs of C57BL/6J Mice Infected with fpt Mutants than in the Lungs of Mice Infected with Parental LVS

The initial host response to respiratory tularemia is characterized by an acute lung infiltration of macrophages and neutrophils in an attempt to clear infection prior to dissemination [39,40,41,42]. This is followed by the development of necrotic foci in the lungs, liver, and spleen, which is one of the hallmark features of end-stage tularemia and indicative of extensive cell death and disseminated disease [43]. Due to the reduced bacterial organ burdens of the *fpt* mutants at day 6 post-infection in mouse lungs, livers, and spleens (Figure 2), we hypothesized that the *fpt* mutants may be attenuated due to a differential host response that is able to clear the *fpt* mutants with reduced resultant pathology. Accordingly, the pathological consequences of infection and the corresponding changes in the murine lung tissue in response to infection with the *fpt* mutants compared to parental LVS were assessed. To measure the extent of pulmonary involvement in the lungs after i.n. inoculation with LVS, LVSΔ*fptA*, or LVSΔ*fptF*, or mock infection with PBS, lungs were harvested at day 6 post-infection (the time point at which there were statistically significant differences in organ burdens in the lung between LVS and the *fpt* mutants, which was maintained for the remainder of the studies). Hematoxylin and eosin (H&E) stained lung tissues were prepared for analysis. The histopathologic parameters examined and the scoring criteria are described in Appendix A. Representative images are shown for the scored slides which highlight the more prominent acute inflammatory changes and increased extent of tissue destruction in the LVS group, and to a lesser extent, the LVSΔ*fptA* group compared to the LVSΔ*fptF* group (Figure 3). LVS caused significant damage by all measures assessed; histopathology was characterized by extensive inflammation, diffuse alveolar destruction, and the presence of neutrophils, macrophages, and lymphocytes (Table 2). Diffuse alveolar destruction due to inflammation caused by neutrophilic, macrophagic, and lymphocytic infiltrates is evident in the LVS-infected lungs (Figure 3A,B). In contrast, infection with LVSΔ*fptA* or LVSΔ*fptF* was characterized by reduced destruction and pulmonary involvement (Table 2). Mice infected with LVSΔ*fptA* had intermediate histopathologic scores that were lower than LVS but higher than LVSΔ*fptF*; compared to LVS, there was reduced inflammation and alveolar destruction and reduced numbers of neutrophils and lymphocytes (Table 2). Despite lower overall scores than LVS, LVSΔ*fptA* induced microscopic manifestations that included some parenchymal destruction due to inflammation, interstitial inflammation in areas of preserved parenchyma, chronic inflammatory infiltrates, and fibrin deposits (Figure 3C,D). Infection with LVSΔ*fptF* induced the lowest histopathology scores in all measures assessed, especially in terms of inflammation, alveolar destruction, fibrin deposition and neutrophilic infiltration (Table 2). Images of lungs infected with LVSΔ*fptF* show less prominent inflammatory manifestations that were limited to interstitial lymphocytic inflammation with hyperplasia of type II pneumocytes, and small foci of inflammation, but unlike the lungs infected with LVS or LVSΔ*fptA*, lung architecture has been preserved (Figure 3E,F). Mock infection did not generate any inflammation or destruction of the lung (Figure 3G,H; Table 2). In summation, mouse lungs infected with the LVSΔ*fptF* mutant showed the least pathology when compared to the levels of tissue destruction caused by LVS, followed by LVSΔ*fptA*. These histopathology results are consistent with their respective levels of attenuation (Table 1) which confirms that virulence is linked to the inflammatory response.

### 2.5. fpt Mutants Induce Altered Proinflammatory Cytokine Responses in the Bronchoalveolar Lavage Fluid of C57BL/6J Mice Compared to Parental LVS

The pathology associated with LVS infection is partly due to the ability of LVS to cause a cytokine storm, which is one of the hallmark features of tularemia [44,45,46]. Because of the reduced lung pathology and inflammatory infiltrates associated with infection with the *fpt* mutants (Figure 3, Table 2), we hypothesized that the *fpt* mutants induce modified cytokine responses in the lungs versus parental LVS. Secreted cytokines (IFN-γ, TNF-α, IL-1β, IL-10, IL-12p70, KC/GRO, IL-6, IL-5, IL-4, and IL-2) in the bronchoalveolar lavage fluid (BALF) were quantified on day 6 post-infection following i.n. inoculation with LVS, LVSΔ*fptA*, or LVSΔ*fptF*, or mock infection with PBS using a 10-plex electrochemiluminescent detection method. Despite some variation between individual mice, BALF samples from LVS- and *fpt* mutant-infected mice showed significantly elevated levels of IFN-γ, TNF-α, IL-1β, IL-12p70, IL-10, IL-2, and KC/GRO compared to mock infection (Figure 4). Levels of IL-4, IL-5, and IL-6 were not significantly elevated in any infection group versus mock infection (Appendix A). There were significant decreases in the levels of secreted IL-1β and IL-10 in the *fpt* mutant-infected BALF samples versus LVS; both *fpt* mutant strains induced significantly less IL-1β and IL-10 secretion in the lungs at day 6 post-infection versus LVS (Figure 4A,C). While not statistically significant, the median levels of IL-2 and IL-12p70 were elevated in the BALF from the *fpt* mutant-infected mice versus LVS (Figure 4B,D). Furthermore, the median level of KC/GRO was reduced in the BALF from *fpt* mutant-infected mice versus LVS (Figure 4G). The levels of IFN-γ and TNF-α were similar between *fpt* mutant-infected and LVS-infected BALF samples (Figure 4E,F). Upon closer analysis of cytokine secretion in individual mice, the mice that produced high levels of IFN-γ had comparably high levels of TNF-α, IL-1β, etc., and the mice that had minimal levels of IFN-γ also had low levels of the other cytokines. The differences in proinflammatory cytokines elicited by the *fpt* mutants versus LVS further support a role for these Fpts in the virulence of *Ft* and the modulation of host immune responses to infection.

### 2.6. fpt Mutants Induce Altered Host Immune Responses in the Lungs of C57BL/6J Mice Compared to Parental LVS

Given the modified cytokine profiles (Figure 4) and histopathological evidence (Figure 3, Table 2) of an altered immune response to the *fpt* mutants, we next hypothesized that early immune cell infiltration to the lungs of *fpt* mutant-infected mice may differ from LVS-infected mice. To interrogate this, immune cell infiltration in the lungs of mice infected with LVS, LVSΔ*fptA*, or LVSΔ*fptF* was evaluated using flow cytometry. Mouse lungs were harvested at day 6 post-infection following i.n. inoculation with LVS, LVSΔ*fptA*, or LVSΔ*fptF*, or mock infection with PBS. Single cell suspensions were processed for analysis using the gating strategy shown in Appendix A. Analysis revealed changes in the overall cell numbers and composition in the LVS- and *fpt* mutant-infected lungs compared to the mock-infected animals. As expected, there was a significant increase in the number of live CD45.2+ immune cells in the LVS- and *fpt* mutant-infected versus mock-infected lungs (Figure 5A). This was mostly due to increased numbers of non-T and non-B cells in the infected lungs (Figure 5D). There was an increase in the number of T cells (Figure 5B) and B cells (Figure 5C) present in the lungs of LVSΔ*fptA*-infected mice, and to a lesser, non-statistically significant extent, in the lungs of LVSΔ*fptF*-infected mice, compared to LVS-infected lungs. Among the T cell subsets, LVS infection caused a decrease in αβ T cells that was not observed with LVSΔ*fptA* or LVSΔ*fptF* infection (Figure 5E). The numbers of γδ T cells and NKT-like cells were not different in LVS- versus mock-infected lungs, but were increased in LVSΔ*fptA*-infected lungs, and the numbers of γδ T cells in the LVSΔ*fptF*-infected lungs were intermediate between the numbers in LVSΔ*fptA*- and LVS-infected lungs (Figure 5F,G).

Among the non-B and non-T cell populations, the numbers of neutrophils (Figure 6B) and non-B Class II+ cells (Figure 6C) were increased in LVS- and *fpt* mutant-infected versus mock-infected lungs; however, while not significant, there was a gradient of neutrophil influx where LVS induced the most neutrophils, followed by LVSΔ*fptA* and LVSΔ*fptF*, which corresponds to the histopathology data (Figure 3, Table 2). The number of NK cells was increased in the LVSΔ*fptA*- and LVSΔ*fptF*-infected lungs compared to both mock- and LVS-infected lungs (Figure 6A). When the non-B Class II+ cell populations were subtyped, LVS failed to induce DC infiltration in contrast to the LVSΔ*fptA* and LVSΔ*fptF* mutants (Figure 6D). The numbers of CD11b+ macrophages (Figure 6E) and Class II+ neutrophils (Figure 6F) were increased in the LVS- and *fpt* mutant-infected lungs versus mock-infected. Additionally, the numbers of CD11b− macrophages were reduced in LVS-infected versus *fpt* mutant-infected lungs (Figure 6G). Finally, when the CD11b− macrophages were further characterized as alveolar macrophages based on the expression of CD68 and CD11c, there were significantly fewer alveolar macrophages present in the LVS-infected lungs versus mock; this decrease did not occur in LVSΔ*fptA*- and LVSΔ*fptF*-infected lungs (Figure 6H). Whereas infection with all strains resulted in significant innate immune responses, differential cell infiltration in response to the *fpt* mutants versus LVS reflects the attenuation of these mutant strains and the induction of immune responses that promote host survival, bacterial clearance, and eventual protection as opposed to the pathological immune response to LVS that results in death of the infected animal.

## 3. Discussion

The 2001 anthrax attacks in the U.S. emphasized the need for the development of diagnostic, preventative, and therapeutic measures against Select Agents of concern for use in biowarfare, including *Ft*. LVS was extensively studied in volunteers where it demonstrated incomplete protection against aerosol challenge with the virulent Type A SchuS4 strain. While great effort has been placed on developing alternative vaccines, it is generally accepted that any new vaccine candidate needs to outperform LVS in terms of safety, immunogenicity, and protective efficacy against aerosolized Type A *Ft* [11,12,14,15,47,48,49,50]. Few LVS-based vaccines have shown protective efficacy against respiratory tularemia caused by heterologous Type A challenge [51,52,53]. Even though there is evidence to suggest that an effective live attenuated tularemia vaccine would likely need to be engineered in the Type A strain, LVS still proves a useful tool to identify targeted mutations that could be generated in the Type A background. The generation of live attenuated bacterial vaccines with defined mutations in metabolic genes has proven to be a successful strategy for vaccine development against *Shigella* and *Salmonella* Typhi, among others [54,55,56,57,58,59], and has been utilized in the generation of vaccine candidates against *Ft*. Several distinct metabolic mutations have been introduced into *Ft* to generate live attenuated, auxotrophic mutants with varying degrees of attenuation and protective efficacies [60,61,62,63,64,65,66,67,68]. Other metabolic targets for attenuation include MFS transporters such as Fpts [34,37]. In this study, we investigated two MFS transporters, FptA and FptF, which have not yet been fully characterized, as attenuating targets for vaccine development.

By all criteria tested, both LVSΔ*fptA* and LVSΔ*fptF* were attenuated for virulence and the host responses to infection with these mutants were substantially modified. The reported LD_50_ of LVS varies depending on route of infection, mouse strain, and methodology [69,70,71]. Using a protocol that incorporates a 50 µL dose volume and deep anesthesia [72,73], the LD_50_ of LVS was determined to be < 60 CFU by the i.n. route in C57BL/6J mice which highlights the extreme virulence of LVS in the mouse model and validates LVS as a useful surrogate model to study the pathogenesis of virulent Type A *Ft*. The low LD_50_ of LVS also emphasizes the attenuation levels of the LVSΔ*fptA* and LVSΔ*fptF* mutant strains; the LD_50_ values of both *fpt* mutants in this model were > 20 times that of LVS. These values supported the evaluation of their ability to confer protection against challenge, and we found that a single inoculation of LVSΔ*fptA* or LVSΔ*fptF*, regardless of the vaccination dose, provided 100% vaccine efficacy against a lethal challenge of ~500 CFU of LVS, which is >8 times the LD_50_.

Infection with *Ft* via the respiratory route begins with bacterial entry into and replication within alveolar macrophages, neutrophils, dendritic cells, monocytes and alveolar type II cells [40,74,75,76] for the first 48-hours followed by dissemination to the liver and spleen [40,41,42], which is consistent with our results. The *fpt* mutant strains were able to colonize the lungs and disseminate to the livers and spleens, initially replicating to the same levels as LVS by day 3 post-infection. However, the *fpt* mutants failed to reach the same peak organ burdens as LVS by day 6 post-infection at which time the numbers of mutant bacteria declined. Furthermore, the *fpt* mutants were almost cleared by day 21 post-infection; clearance is an important consideration for live attenuated vaccine studies, as a strain that is unable to be cleared would not be a viable vaccine candidate.

The host initially responds to *Ft* with an acute lung infiltration of macrophages and neutrophils, which become infected shortly after exposure [39,40,41,42]. During severe respiratory tularemia, the alveolar spaces become filled with neutrophils, macrophages, necrotic cellular debris, and bacteria [42], which may contribute to the resultant severe pneumonia and pathology. However, this acute immune response is insufficient to clear *Ft* from the lungs and prevent dissemination, and instead, contributes to pathogenesis; uncontrolled bacterial replication and neutrophil infiltration contribute to the extensive tissue destruction and the unrestrained production of proinflammatory cytokines in what is known as the cytokine storm, which is associated with significant pathology, extensive organ damage, multiple organ failure, the development of severe sepsis, and eventual host death [38,39,45,71,77,78,79,80,81,82,83]. This response is consistent with our organ burden, histopathology and flow cytometry data showing increasing organ burdens (or the failure to control replication and dissemination) and extensive inflammation, diffuse alveolar destruction, and the presence of neutrophils, macrophages, and lymphocytes in LVS-infected lungs on day 6 post infection. However, *fpt* mutant-infected lungs showed reduced pathology and altered host responses characterized by reduced inflammation, including reduced neutrophil infiltration, and reduced alveolar destruction. This evidence suggests that modified host responses to the *fpt* mutant strains occurred that (1) controlled the bacterial burden as demonstrated by the reduced organ burdens at day 6 post-infection, (2) induced minimal tissue pathology and damage, which reduced the severity of disease, and (3) promoted the survival of mice infected with *fpt* mutants.

The pathology associated with the cytokine storm during LVS infection is one of the hallmark features of tularemia [44,45,46,84]. In good agreement with prior reports [85], we observed that *Ft* induced high levels of the potent proinflammatory cytokines IFN-γ, TNF-α, IL-1β, and KC/GRO. Mouse knockout models have demonstrated critical roles for IFN-γ and TNF-α in the immune response against *Ft* as knockout mice succumb to tularemia more rapidly than wildtype mice [86,87,88,89,90,91,92,93]. It is therefore speculated that elevated IFN-γ and TNF-α are necessary but not sufficient for survival against *Ft* infection. Encouragingly, we found that the *fpt* mutants induced similarly elevated levels of IFN-γ and TNF-α in the BALF as LVS, which may contribute to the development of protective immune responses in *fpt* mutant-infected mice that are not seen in the LVS-infected mice that succumb by day 10 post-infection, which is not sufficient time for the development of the adaptive response. KC/GRO is a marker of the severe sepsis associated with tularemia [78] which was correspondingly elevated in the LVS-infected BALF. However, the *fpt* mutants induced reduced levels of KC/GRO in the BALF versus LVS, suggesting a reduction of severe sepsis in the *fpt* mutant-infected mice and providing an additional measure of their attenuation. The role for IL-1β in the defense against *Ft* is not yet well understood; however, IL-1β in the BALF of C57BL/6J mice i.n. infected with a lethal dose of LVS is higher at day 3 post-infection than at day 6 post-infection [94]. Therefore, one could speculate that early IL-1β secretion is insufficient to promote bacterial clearance and host survival. Others have shown that hypersecretion of IL-1β in mice infected with attenuated mutant strains of LVS was protective and associated with survival [95]; we observed the opposite phenomenon wherein our *fpt* mutant strains induced significantly less IL-1β than LVS, a difference that highlights the incomplete understanding of the role of IL-1β in the response against LVS. Furthermore, LVS induced the production of the anti-inflammatory cytokine IL-10 which has been shown to be produced by activated lung DCs early in the course of respiratory tularemia [74,80,84] and by regulatory T cells (Tregs) [96] which are present in the lungs as soon as one day post-infection [80]. It seems confounding that a pathogen capable of producing a cytokine storm would also induce anti-inflammatory mediators. However, the timeline of cytokine secretion is an important factor; it has been speculated that *Ft* induces IL-10 early during the course of respiratory tularemia to dampen or delay the initial pro-inflammatory cytokine response [84], which may explain the subsequent reduction in IL-1β secretion, facilitating bacterial growth and survival [80]. We found that the *fpt* mutants failed to induce comparably elevated levels of IL-1β and IL-10 as LVS. As IL-1β is released from immune cells upon pyroptosis, it is possible that the reduced histopathology in the *fpt* mutant-infected lungs can be attributed to decreased pyroptosis and IL-1β secretion compared to LVS. Additionally, the reduced levels of IL-10 in the *fpt* mutant-infected lungs may be indicative of the *fpt* mutants’ inability to invoke an anti-inflammatory cytokine milieu, as observed during LVS infection, providing further evidence that FptA and FptF are important factors required for the pathogenesis of *Ft*. These differences in pro- and anti-inflammatory cytokines elicited in response to the *fpt* mutants versus LVS underscore the balance between pathological inflammation (as seen in LVS-infected mice) and protective inflammation (as seen in *fpt* mutant-infected mice) and support the contribution of the *fptA* and *fptF* gene products in the modulation of the host inflammatory response by LVS.

In addition to reduced lung histopathology and altered cytokine responses in the BALF of *fpt* mutant-infected mice, immune cell infiltration in lungs of *fpt* mutant-infected mice was modified compared to LVS. LVS infection was characterized by an increase in non-T and non-B cells, specifically neutrophils and non-B Class II+ cells (including CD11b+ macrophages and Class II+ neutrophils), with a concomitant decrease in T cells and alveolar macrophages compared to mock-infected lungs. Others have reported similar innate immune cell profiles in a murine model of respiratory tularemia [40]. Mature neutrophils typically do not express Class II, but a novel phenomenon in which granulocytes (including neutrophils) acquire antigen presenting capabilities that influence T cell responses has been reported [97,98,99]. A significant proportion of the neutrophil population was Class II+ in the LVS- and *fpt* mutant-infected lungs and may represent the first evidence of this phenomenon occurring during *Ft* infection. Generally, responses to the *fpt* mutant strains, especially the LVSΔ*fptA* mutant strain, and to a lesser extent, the LVSΔ*fptF* mutant strain, were characterized by increases in B cells, T cells, NK cells, DCs, and CD11b− macrophages versus LVS. It has been shown that strong and early protection against sublethal doses of LVS is dependent on B cells through limiting bacterial growth early after infection, and this protective mechanism is dependent on IFN-γ [100]. Therefore, it is possible that the increased presence of B cells in the *fpt* mutant-infected lungs on day 6 post-infection may contribute to the observed reduction in organ burdens among these mice at this time point. Furthermore, the increased presence of DCs in the *fpt* mutant-infected lungs may also contribute to the reduction of *fpt* mutant organ burdens as DCs have been shown to help limit early pathogen replication [101]. The increased levels of IL-10 in the LVS-infected BALF may underlie the reduced levels of T cells in the LVS-infected lungs; IL-10 suppresses T cell proliferation. Conversely, the reduced levels of IL-10 in the *fpt* mutant-infected BALF versus LVS may explain the increased presence of T cells in the *fpt* mutant-infected lungs. Further studies are needed to identify the source of IL-10 in the murine lung during respiratory tularemia. There is evidence of NK regulation of T cell activity during respiratory tularemia [102]. Additionally, it is thought that another role for NK cells in respiratory tularemia is to lyse infected alveolar macrophages [102]. We speculate that the severe lung pathology and reduced population of alveolar macrophages in the LVS-infected lungs may be due in part to lysis of these macrophages, perhaps by NK cells, which have become heavily infected with *Ft*. It has also been shown that γδ T cells can directly activate NK cells [103], and therefore the increase in NK cells in the *fpt* mutant-infected lungs versus LVS- and mock-infected lungs may be reflective of the increased presence of γδ T cells in the *fpt* mutant-infected lungs. Taken together, B cells in *fpt* mutant-infected mice may be receiving more T cell help as a result of increased antigen presentation by DCs and higher NK cell activity in mutant-infected lungs. The differential host responses to the *fpt* mutants reflect their levels of attenuation and may indicate a protective response, perhaps due to less pathological inflammation. Additional studies at later time points are needed to identify the implications of these early responses on the development of protective immunity against subsequent challenge.

This highly sensitive murine model of respiratory tularemia allowed the dissection of many parameters of the host pulmonary responses to LVS and to the *fpt* mutants following i.n. inoculation. In agreement with the literature, LVS induced a highly pathogenic immune response that contributes to the lethality of LVS in mice. The differences between the histopathology, BALF cytokines, and the immune cell composition between LVS- and *fpt* mutant-infected lungs allowed us to evaluate the contribution of FptA and FptF to the pathogenesis of *Ft*. We speculate that the respective (yet unknown) substrates of the FptA and FptF transporters are essential for *Ft* to reach a critical threshold of bacterial burden in the host to facilitate full virulence and pathogenesis. Alternatively, the *fpt* mutants may be impaired such that they are unable to induce the required virulence gene profiles for pathogenesis and are instead cleared through a more protective and less pathological immune response by the host. Further studies involving the analysis of the in vivo transcriptomic profiles of the *fpt* mutants versus LVS are needed to identify the specific effects of deleting the *fpt* gene products on virulence gene expression. Ultimately, the differential host responses to the *fpt* mutant strains versus to LVS support the essential roles for FptA and FptF in the modulation of the host immune response such that LVS can establish conditions to permit bacterial replication, spread, pathogenesis and eventual host death. These data support the further interrogation of *fptA* and *fptF* genes in the virulent Type A SchuS4 strain towards the ultimate goals of developing live attenuated tularemia vaccines and understanding the contributions of Fpts in the pathogenesis of *Ft*.

## 4. Materials and Methods

### 4.1. Bacteria and Growth Conditions

*Ft* LVS, which was kindly provided by Dr. Karen Elkins (CBER/FDA, Rockville, MD, USA), LVSΔ*fptA* and LVSΔ*fptF* strains [34] were preserved at −80 °C in Mueller-Hinton Broth (MHB) (BD, Sparks, MD, USA) with 15% glycerol added. Complete MHB includes 1% IsoVitaleX (BD, Sparks, MD, USA), 0.1% glucose (Sigma-Aldrich, St. Louis, MO, USA), and 0.25% ferric pyrophosphate (Sigma-Aldrich, St. Louis, MO, USA) and was used for all liquid cultures. Mueller-Hinton agar (MHA) (BD, Sparks, MD, USA) was used for solid cultures and was augmented as defined above but also contained 10% defibrinated sheep blood (Lampire Biological Laboratories, Pipersville, PA, USA). Growth on solid media was performed at 37 °C, 5% CO_2_.

### 4.2. Mice

Wildtype C57BL/6J mice were purchased from The Jackson Laboratory (The Jackson Laboratory, Bar Harbor, ME, USA) for use in i.n. attenuation, organ burden enumeration, and protective capacity studies. For Appendix A; wildtype BALB/c mice were purchased from Charles River Laboratories (Charles River Laboratories, Frederick, MD, USA) for use in i.p. attenuation studies. Infected mice were housed in the University of Maryland animal biohazard safety level 2 (ABSL-2) facility for the duration of the studies. All animal experiments were conducted according to protocols approved by the UMB Institutional Animal Care and Use Committee (IACUC).

### 4.3. Attenuation of fpt Mutant Strains in Mice

Six- to eight-week-old female BALB/c mice (Charles River Laboratories, Frederick, MD, USA) were initially used to test the *fpt* mutant strains in vivo for attenuation. Groups of 4 BALB/c mice per group were injected i.p. with either LVS or the *fpt* mutant, or mock-infected with PBS. To reassess the attenuation levels of *fpt* mutants in a more stringent model of protection against lethal *Ft* following vaccination, groups of 4 or 6 eight- to nine-week-old male and female C57BL/6J mice per dosage concentration were anesthetized i.p. with a cocktail of 90 mg/kg of ketamine and 9 mg/kg of xylazine and inoculated i.n. with either LVS or the *fpt* mutant suspended in 50 µL of PBS, or mock-infected with PBS. Studies utilizing dosage groups near the LD_50_ values were repeated to confirm the LD_50_ values. Mice were monitored daily for survival and clinical signs of infection (weight loss, lethargy, and ruffling of fur), and weights were monitored every other day for 28 days post-infection. Mice were assigned health assessment scores based on general physical appearance (skin, fur, etc.), ambulation, potential dehydration, response to stimuli, movement, evidence of food consumption (feces in bedding), grooming and any clinical sign that may indicate disease or illness (lethargy, hunched posture, head tucked into abdomen, rough hair, visible tumors, etc.). Mouse health was scored based on the following criteria: condition 1 for normal activity, where mice are bright, alert, reactive, healthy, shiny, have normal posture and lack dehydration; condition 2 for mice that are quiet, alert, reactive with early stage piloerection, normal posture and mild dehydration; condition 3 for mice that are quiet and not very reactive, and exhibit mild piloerection, dull coats, hunched posture, difficult ambulation, moderate dehydration, and scruff remains tented (lifted) for 2–3 s; and condition 4 for mice that are non-reactive with severe piloerection, dull and dirty coats, hunched and squinting, severely dehydrated, tented scruff that when lifted does not return, severe weight loss (>20%) and respiratory distress. Mice reaching a clinical score of 4 or losing >20% of their body weight were euthanized as required by the UMB IACUC.

### 4.4. Protective Capacity of fpt Mutant Strains in Mice

The surviving mice from the “Attenuation of *fpt* Mutant Strains in Mice” studies listed above were utilized for challenge studies to determine protective capacity of the *fpt* mutant strains. Twenty-eight days post-vaccination, the surviving mice were anesthetized and inoculated i.n. with ~500 CFU of LVS suspended in 50 µL of PBS. After challenge, all mice were monitored daily for clinical signs and weights were monitored every other day as described above for 21 days post-challenge. Vaccine efficacy was calculated using the following formula: (Attack rate unvaccinated − Attack rate vaccinated) ÷ (Attack rate unvaccinated) × 100.

### 4.5. Quantification of Bacterial Organ Burdens in Mice

To determine the bacterial organ burdens of *fpt* mutant strains in vivo, groups of 4 eight-week-old male and female C57BL/6J mice per strain per time point were anesthetized and inoculated i.n. with ~350 CFU of either LVS or the *fpt* mutant suspended in 50 µL of PBS. Mice were euthanized at the designated time points post-infection, and organs were harvested to enumerate bacterial load. Organs were halved for use in quantification of bacteria organ burdens and histopathology analysis. Lungs, livers, and spleens from infected mice were homogenized using the BeadBug Microtube Homogenizer (Benchmark Scientific, Sayreville, NJ, USA) according to manufacturer’s protocol in PBS. The bacterial load was enumerated by serial plating on MHA plates.

### 4.6. Histopathology Analysis of Infected Mouse Lungs

To assess the pathological consequences and changes in lung tissue in response to infection with the *fpt* mutants, groups of 4 eight-week-old male and female C57BL/6J mice were anesthetized and inoculated i.n. with ~350 CFU of either LVS or the *fpt* mutant suspended in 50 µL of PBS. Mice were euthanized at day 6 post-infection and lungs were harvested for histopathology analysis. Lungs were fixed in 10% Formalin, neutral buffered (Sigma-Aldrich, St. Louis, MO, USA) for a minimum of 24 h. After fixation, the tissues were processed and embedded in paraffin. Tissue blocks were sectioned at 5 µm and stained with hematoxylin and eosin by the University of Maryland School of Medicine Histology Core Laboratory. Stained tissues were examined on a Aperio ScanScope CS (Aperio, Vista, CA, USA) and assessed by two pathologists. Both observers were blinded regarding the type of infection and performed their scoring independently. The histopathologic parameters examined and scoring criteria are described in Appendix A.

### 4.7. Quantification of Secreted Cytokines in Murine Bronchoalveolar Lavage Fluid

To measure the secreted cytokines in the BALF in response to infection with the *fpt* mutants, LVS, or mock infection with PBS, groups of 5 eight-week-old male and female C57BL/6J mice were anesthetized and inoculated i.n. with ~350 CFU of either LVS or the *fpt* mutant suspended in 50 µL of PBS, or mock-infected with PBS. Mice were euthanized on day 6 post-infection by i.p. anesthetic overdose (cocktail of 360 mg/kg of ketamine and 36 mg/kg of xylazine) followed by secondary exsanguination. Bronchoalveolar lavage fluid was harvested in 1 mL of sterile PBS via an intratracheal catheter attached to a 1mL syringe, and BSA (Reagent Diluent Concentrate 2, R&D Systems, Minneapolis, MN, USA) was added for a final concentration of 1%. BALFs were frozen at −80 °C until cytokines were analyzed using the V-PLEX Proinflammatory Panel 1 Mouse Kit assay based on an electrochemiluminescent (ECL) detection method (Meso Scale Diagnostics, LLC, Rockville, MD, USA) according to the manufacturer’s recommendations. Data were collected using the MESO QUICKPLEX SQ 120 instrument and analyzed with DISCOVERY WORKBENCH^®^ V4.0 data analysis software (Meso Scale Diagnostics, LLC, Rockville, MD, USA). Briefly, samples were thawed on the day of analysis and standards and BALF samples (50 µL) were added to appropriate wells and incubated for two hours at room temperature with shaking (700 rpm). The fluid was then removed, and the wells were washed 3 times with wash buffer (0.05 % Tween-20 in PBS). Detection antibodies were added to each well and incubated for 2 h at room temperature with shaking, followed by washing 3 times. After washing, 150 µL of 2× reading buffer was added to each well. The plate was analyzed on the MSD instrument immediately. For the purposes of statistical analyses, any value that was below the lowest limit of detection (LLOD) for the assay was replaced with half of LLOD of the assay.

### 4.8. Characterization of the Immune Response in Infected Mouse Lungs

To quantify the responding immune cell populations after infection with the *fpt* mutants, LVS or mock infection with PBS, groups of 5 eight-week-old male and female C57BL/6J mice were anesthetized and inoculated i.n. with ~350 CFU of either LVS or the *fpt* mutant suspended in 50 µL of PBS, or mock-infected with PBS. Mice were euthanized on day 6 post-infection by i.p. anesthetic overdose (cocktail of 360 mg/kg of ketamine and 36 mg/kg of xylazine) followed by secondary exsanguination. Pulmonary perfusion was performed with sterile PBS prior to harvesting the lungs. Once harvested, the lungs were immersed in cold cell staining buffer (BioLegend, San Diego, CA, USA) on ice until processing. Lungs were minced using autoclaved scissors in 15 mL of digestion solution containing 0.525 mg/mL Collagenase D (Roche Diagnostics, Mannheim, Germany) and 0.06 mg/mL DNase I (Roche Diagnostics, Mannheim, Germany) in RPMI 1640 + L-glutamine (Gibco, Gaithersburg, MD, USA) supplemented with 1% Penicillin/Streptomycin (Gibco, Gaithersburg, MD, USA) and 10% heat inactivated/endotoxin free FBS (Gemini Bio, West Sacramento, CA, USA), then filter sterilized using a 0.22 μm filter. Minced lungs were incubated in the digestion solution at 37 °C for 90 min with shaking and filtered through a 70 μm nylon mesh cell strainer (Corning, Glendale, AZ, USA). RPMI 1641 + L-glutamine was used to wash the strainer. Cell suspensions were centrifuged at 1300 rpm, at 22 °C, for 10 min. Supernatants were carefully removed and washed once with room temperature RPMI 1641 + L-glutamine. The cell pellets were each resuspended in 5 mL of 1.075 density Percoll (Sigma-Aldrich, St. Louis, MO, USA) before 5 mL of 1.03 density Percoll was carefully layered onto the cell suspensions and centrifuged at 1000 rpm, without brake, at 4 °C for 20 min. The interfaces containing the cells were washed twice in room temperature RPMI 1641 + L-glutamine before being resuspended in 1 mL of cold cell staining buffer and placed on ice. Small aliquots of cells were diluted in 0.04% Trypan Blue Solution (Sigma-Aldrich, St. Louis, MO, USA) and cell counts were determined using the Countess Automated Cell Counter (Invitrogen, Carlsbad, CA, USA). An aliquot of each sample containing 1 × 10^6^ cells in 100 µL was incubated in TruStain FcX Plus (BioLegend, San Diego, CA, USA) according to the manufacturer’s protocol to block Fc receptors before being stained with 50 µL of the following fluorochrome-conjugated antibody cocktail according to the manufacturer’s protocol (BioLegend, San Diego, CA, USA): F4/80 (BV421, CAT#123137), Viability (Zombie NIR CAT#423106), CD11b (BV570 CAT#101233), NK1.1 (BV605 CAT#108753), CD3 (BV711 CAT#100349), I-A/I-E also known as Class II (BV785 CAT#107645), Ly6C (FITC CAT#128005), TCRgd (PE CAT#118107), CD11c (PE/Dazzle594 CAT#117347), Ly6G (PerCP CAT#127653), CD68 (PE-Cy7 CAT#137015), CD45.2 (AF700 CAT#109821), TCRb (APC-Fire750 CAT#109245), B220 (APC-Fire810 CAT#103277), and True-Stain Monocyte Blocker (CAT#156603). Cells were washed and resuspended in 200 µL of eBioscience IC Fixation Buffer (Invitrogen, Carlsbad, CA, USA) and incubated on ice for 20 min. Following fixation, cells were washed in 200 µL of cold cell staining buffer and then resuspended in 200 µL of cold cell staining buffer and used for flow cytometry analysis. Cells were analyzed on an Aurora Spectral Cytometer (Cytek Biosciences, Bethesda, MD, USA). All data were analyzed using FlowJo (FlowJo LLC, Ashland, OR, USA). Complete gating strategy can be found in Appendix A. Briefly, samples were first gated on FSC-A and SSC-A to collect cells, then gated on Zombie NIR− cells to identify live cell populations. To discriminate between immune cells and other remaining lung cell populations, CD45.2+ cells were gated for further analysis. T cells were defined as CD3+ B220− populations. T cells were further subtyped into gamma-delta T cells (γδTCR+TCRβ−), alpha-beta T cells (NK1.1−, TCBβ+), and natural killer-like T (NKT-like) cells (NK1.1+TCRβ+). B cells were defined as CD3− B220+ populations. Non-B and non-T cells were defined as CD3− B220− populations. From the non-B and non-T cell population, natural killer (NK) cells were gated as NK1.1+ I-A/I-E−. Neutrophils were gated from non-B and T cell populations as Ly6G+ CD11b+ cells. Non-B Class II+ cells were gated from non-B and non-T cell populations as I-A/I-E+ cells. From the non-B Class II+ cells, the following populations were defined: dendritic cells (F4/80−, CD11b−), CD11b− macrophages (F4/80+, CD11b−), CD11b+ macrophages (F4/80+, CD11b+), and Class II+ neutrophils (F4/80−, CD11b+). The Class II+ neutrophil population was confirmed by the expression of Ly6G as well as Ly6C. The CD11b− macrophages were further subtyped into alveolar macrophages based on expression of CD68 and CD11c. Frequencies of cell populations were calculated based on the gating strategy described above and absolute number of each cell population was determined as follows: the number of immune cells were calculated by multiplying the frequency of live CD45.2+ cells by total live cell number from cell isolation. The number of T cells, B cells and non-T/B cells were calculated by multiplying the frequency of the corresponding population of cells by the number of live immune cells. The number of αβ T cells, γδ T cells, and NKT-like cells were calculated by multiplying the frequency of the corresponding population of cells by the total number of T cells. The number of NK cells, neutrophils, and non-B Class II+ cells were calculated by multiplying the frequency of the corresponding population of cells by the total number of non-T/B cells. The number of DCs, CD11b+ macrophages, Class II+ neutrophils and CD11b− macrophages were calculated by multiplying the frequency of the corresponding population of cells by the total number of non-B Class II+ cells. The number of alveolar macrophages were calculated by dividing the frequency of alveolar macrophages by the total number of CD11b− macrophages.

### 4.9. Statistical Analysis

A two-tailed *t* test was used to determine statistical significance in organ bacterial burden experiments. Results were considered statistically significant at a *p* value of <0.05. A two-tailed *t* test was used to determine statistical significance in cytokine secretion in BALF samples. Results were considered statistically significant at a *p* value of <0.05. A one-way ANOVA with a Tukey’s post-test was used to determine statistical significance in the flow cytometric analyses. Results were considered statistically significant as follows: ****, *p* < 0.0001; ***, *p* = 0.001; **, *p* < 0.001; *, *p* < 0.05. Data analysis was performed using GraphPad Prism 9.0 (GraphPad Software, Inc., San Diego, CA, USA).

## Figures and Tables

**Figure 1 pathogens-10-00799-f001:**
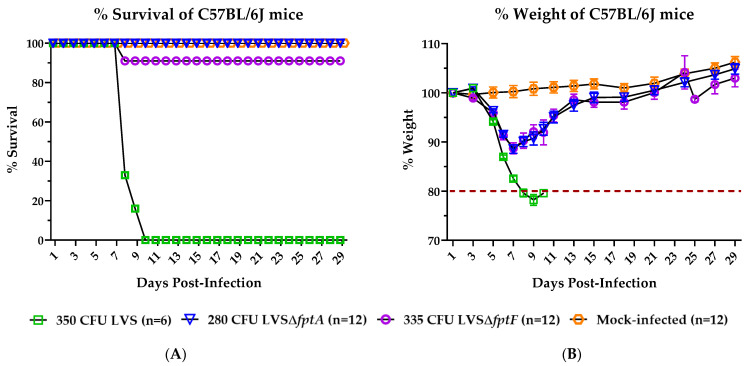
Attenuation of *fpt* mutant strains in the murine model of respiratory tularemia. The survival rates (**A**) and percentage of initial weight (**B**) of eight- to nine-week-old male and female C57BL/6J mice following infection with similar doses of LVS or LVSΔ*fpt* mutant strains were followed for 28 days post-infection. Groups of 6 mice were inoculated i.n. with ~350 CFU of either LVS, LVSΔ*fptA* or LVSΔ*fptF*, or PBS. % weights are graphed as means with SEM. Mice were euthanized once they lost > 20% of initial starting weight.

**Figure 2 pathogens-10-00799-f002:**
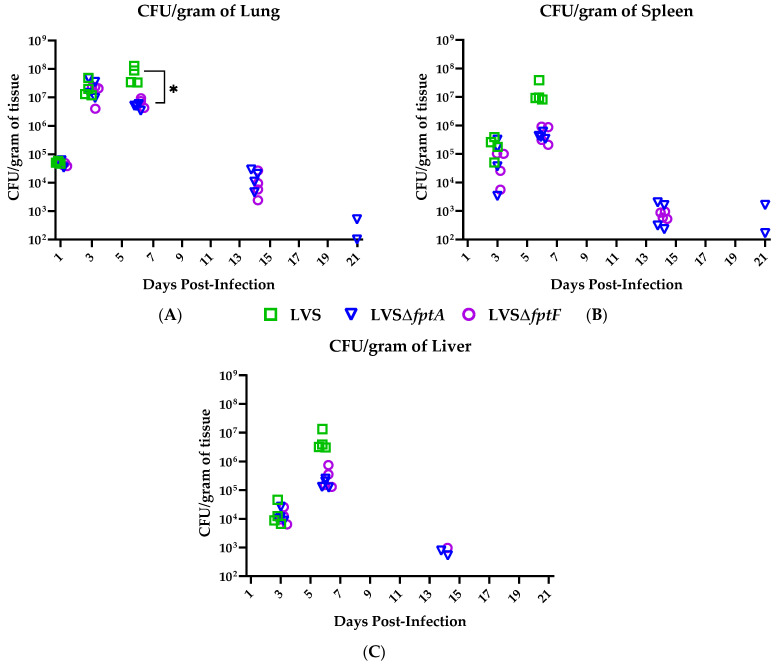
Colonization and dissemination of *fpt* mutant strains in the murine model of respiratory tularemia. The bacterial organ burdens in the lungs (**A**), spleens (**B**) and livers (**C**) of eight-week-old male and female C57BL/6J mice following infection with LVS or LVSΔ*fpt* mutant strains were enumerated. Groups of 4 mice per strain per time point were inoculated i.n. with ~350 CFU of either LVS, LVSΔ*fptA* or LVSΔ*fptF* and organs were harvested for bacterial enumeration on days 1, 3, 6, 14, and 21 post-infection. Symbols indicate burdens in individual mice. *, *p* < 0.05 by a two-tailed *t* test. A lack of symbols at any of the designated time points of harvest indicates an absence of bacteria in these organs. The limit of detection was 10^2^ CFU/gram of tissue. All *fpt* strains had statistically significant lower bacterial burdens than LVS in mouse lungs on day 6 (*p* < 0.05).

**Figure 3 pathogens-10-00799-f003:**
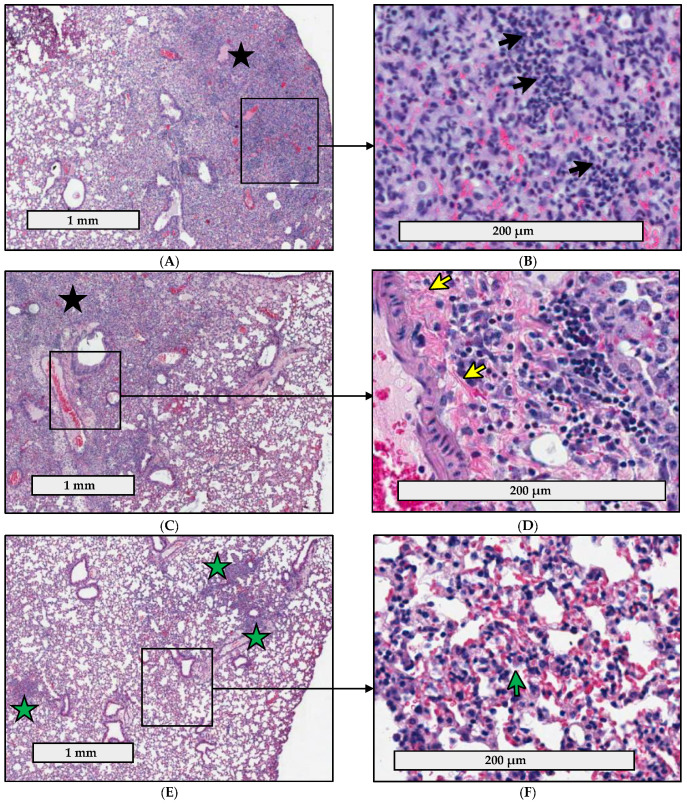
Histopathologic changes in the lungs of mice infected with *fpt* mutant strains. Groups of 4 eight-week-old male and female C57BL/6J mice were inoculated i.n. with ~350 CFU of either LVS (**A**,**B**); LVSΔ*fptA* (**C**,**D**); LVSΔ*fptF* (**E**,**F**); or PBS (**G**,**H**). Representative photomicrographs of H&E-stained lung sections are shown. Rulers in (**A**,**C**,**E**,**G**) denote 1 mm, and in (**B**,**D**,**F**,**H**) denote 200 µm. Prominent inflammatory changes were observed in LVS-infected lungs with diffuse alveolar destruction (**A**, black asterisk) caused by inflammatory cell infiltrates (**B**) including neutrophils (black arrows), lymphocytes and macrophages. LVSΔ*fptA* induced similar microscopic manifestations (**C**) including parenchymal destruction (black asterisk) and interstitial inflammation in areas of preserved parenchyma (**C**, right). (**D**) shows area of chronic inflammatory infiltrate (**D**, right) as well as fibrin deposition (yellow arrow) after LVSΔ*fptA* infection. (**E**,**F**) show the less prominent inflammatory manifestations in LVSΔ*fptF*-infected lungs which were mainly limited to small foci of inflammation (green asterisks) with preserved overall architecture (**E**) as well as interstitial inflammation (mainly lymphocytic) with hyperplasia of type II pneumocytes (**F**, green arrow). (**G**,**H**) show normal lung histology in mock-infected mice (negative control).

**Figure 4 pathogens-10-00799-f004:**
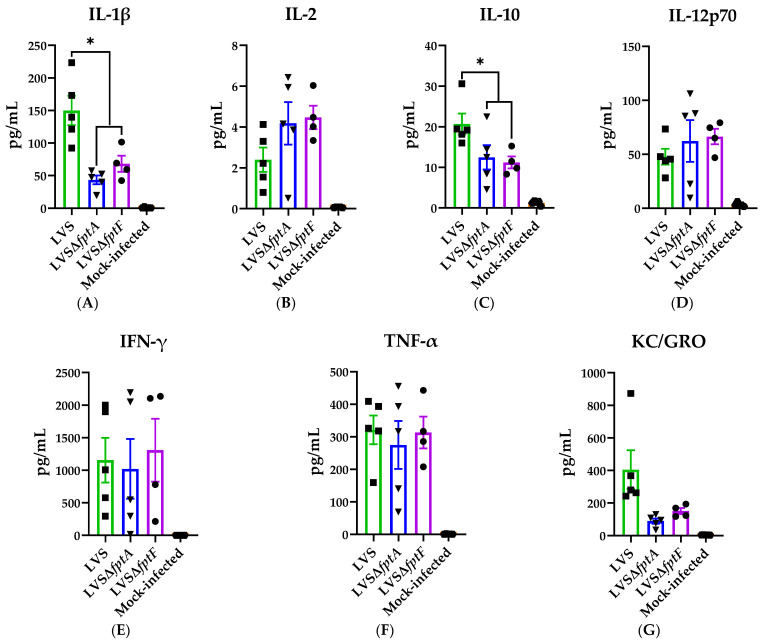
Cytokine secretion in the bronchoalveolar lavage fluid (BALF) of mice infected with *fpt* mutant strains. Groups of 5 eight-week-old male and female C57BL/6J mice were inoculated i.n. with ~350 CFU of either LVS, LVSΔ*fpt*A, LVSΔ*fptF*, or PBS. BALF was harvested at day 6 post-infection for measurement of secreted cytokines using MSD. Symbols indicate cytokine values in individual mice. Bars represent means with SEM from duplicate measurements from one experiment. *, *p* < 0.05 by a two-tailed *t* test. All treatment samples were statistically significantly upregulated versus the mock-infected control. One BALF sample from the LVSΔ*fptF* group was excluded from analyses due to inefficient BALF harvest.

**Figure 5 pathogens-10-00799-f005:**
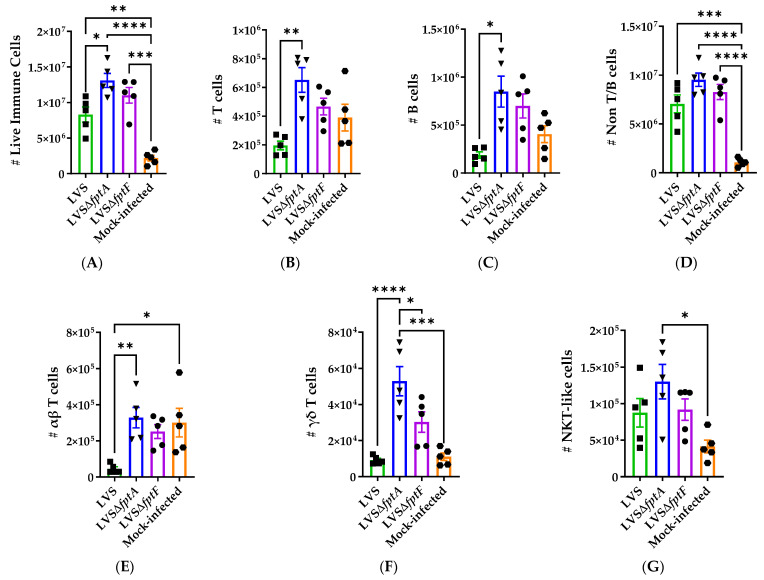
The numbers of T cells, B cells and non-T/B cells change during infection of mice with *fpt* mutants. Groups of 5 eight-week-old male and female C57BL/6J mice were inoculated i.n. with ~350 CFU of either LVS, LVSΔ*fptA*, or LVSΔ*fptF*, or PBS. Lungs were harvested at day 6 post-infection and stained for flow cytometric analysis of responding immune cell populations. Symbols indicate values from individual mice. Bars represent means with SEM. ****, *p* < 0.0001; ***, *p* = 0.001; **, *p* < 0.001; *, *p* < 0.05 by a one-way ANOVA with a Tukey’s post-test. The number of live immune cells (**A**) was calculated by multiplying the frequency of live CD45.2+ cells (Appendix A) by total live cell number from cell isolation. The numbers of T cells (**B**), B cells (**C**) and non-T/B cells (**D**) were calculated by multiplying the frequency of the corresponding population of cells (Appendix A) by the number of live immune cells (**A**). The numbers of αβ T cells (**E**), γδ T cells (**F**), and NKT-like cells (**G**) were calculated by multiplying the frequency of the corresponding population of cells (Appendix A) by the total number of T cells (**B**).

**Figure 6 pathogens-10-00799-f006:**
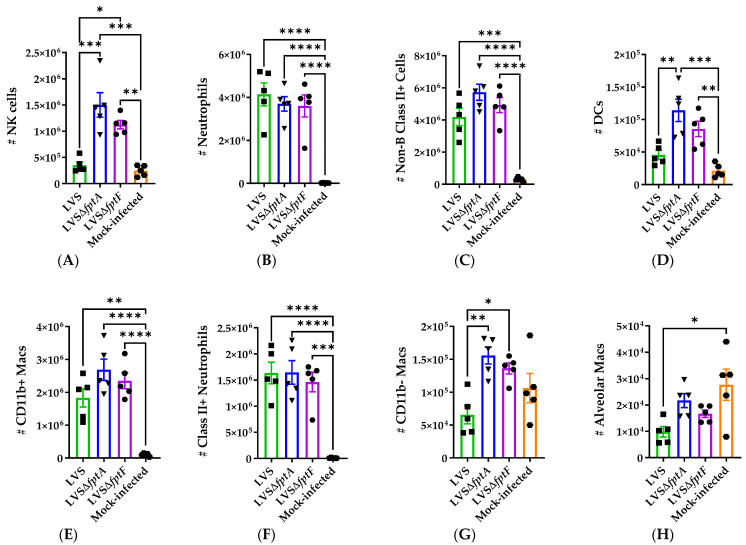
The numbers of subsets of non-T/B cells change during infection of mice with *fpt* mutants. Groups of 5 eight-week-old male and female C57BL/6J mice were inoculated i.n. with ~350 CFU of either LVS, LVSΔ*fptA*, or LVSΔ*fptF*, or PBS. Lungs were harvested at day 6 post-infection and stained for flow cytometric analysis of responding immune cell populations. Symbols indicate values from individual mice. Bars represent means with SEM. ****, *p* < 0.0001; ***, *p* = 0.001; **, *p* < 0.001; *, *p* < 0.05 by a one-way ANOVA with a Tukey’s post-test. The numbers of NK cells (**A**), Neutrophils (**B**), and non-B Class II+ cells (**C**) were calculated by multiplying the frequency of the corresponding population of cells (Appendix A) by the total number of non-T/B cells (Figure 5D). The numbers of DCs (**D**), CD11b+ macrophages (**E**), Class II+ Neutrophils (**F**) and CD11b− macrophages (**G**) were calculated by multiplying the frequency of the corresponding population of cells (Appendix A) by the total number of non-B Class II+ cells (**C**). The number of alveolar macrophages (**H**) was calculated by dividing the frequency of alveolar macrophages (Appendix A) by the total number of CD11b− macrophages (**G**).

**Table 1 pathogens-10-00799-t001:** Attenuation and protective capacity of *fpt* mutant strains in the murine model of respiratory tularemia.

Strain	Vaccination Dose (i.n.) on Day 1 (CFU)	Survival Post-Vaccination on Day 28	LD_50_	Survival Post-Challenge with ~500 CFU LVS (i.n.) on Day 29 ^1^	Vaccine Efficacy ^2^	Sterilizing Immunity ^3^
**PBS**	-	24/24		1/24	-	-
**LVS**	60	1/6	<60 CFU	1/1	100%	Yes
	121	0/6		-	-	-
	175	1/6		1/1	100%	Yes
	242	0/6		-	-	-
	350	0/6		-	-	-
	483	0/6		-	-	-
	633	0/6		-	-	-
**LVSΔ*fptA***	~280	12/12	~1400 CFU	12/12	100%	Yes
	~560	10/12		10/10	100%	-
	~1120	5/11 ^4^		5/5	100%	-
	~1680	7/11 ^5^		7/7	100%	Yes
	3250	0/4		-	-	-
	32,500	0/4		-	-	-
**LVSΔ*fptF***	~335	11/12	~2010–5167 CFU	11/11	100%	Yes
	~670	9/9 ^6^		9/9	100%	-
	~1340	11/12		11/11	100%	-
	~2010	8/12		8/8	100%	Yes
	5167	0/4		-	-	-
	51,667	0/4		-	-	-

^1^ Surviving mice were challenged i.n. with ~500 CFU of LVS on day 29 post-infection and followed for another 21 days post-challenge. ^2^ Vaccine efficacy was calculated using the following formula: (Attack rate unvaccinated − Attack rate vaccinated) ÷ (Attack rate unvaccinated) × 100. ^3^ Sterilizing immunity was determined by lung burden enumeration 21 days post-challenge. ^4^ One mouse was removed from analysis in the LVSΔ*fptA* group receiving ~1120 CFU due to inefficient dose delivery. ^5^ One mouse for the intended LVSΔ*fptA* group receiving ~1680 CFU died in the animal facility for unknown reasons prior to inoculation. ^6^ Three mice were removed from analysis in the LVSΔ*fptF* group receiving ~670 CFU because these mice did not have adequate access to water and exhibited severe weight loss due to dehydration.

**Table 2 pathogens-10-00799-t002:** Scoring results of histpoathology analysis ^1^.

Strain	LVS	LVSΔ*fptA*	LVSΔ*fptF*	Mock-Infected
**(1a) Global Extent of Inflammation**	1.75	1.5	1.25	0.0
**(1b) Surface Area of Alveolar Destruction**	1.5	1.25	0.25	0.0
**(1c) Interstitial Involvement**	0.75	0.75	0.5	0.0
**(1d) Foci of Inflammation**	1.0	0.75	0.5	0.0
**(2a) Neutrophils**	1.0	0.5	0.0	0.0
**(2b) Macrophages**	0.75	0.75	0.5	0.0
**(2c) Lymphocytes**	1.0	0.75	0.75	0.0
**(3a) Fibrin Deposition**	0.75	0.75	0.0	0.0
**(3b) Hyperplasia of Type II Pneumocytes**	1.0	0.75	0.75	0.0
**Cummulative Score**	9.5	7.75	4.5	0.0

^1^ Mean scores are shown for groups of 4 eight-week-old male and female C57BL/6J mice inoculated i.n. with ~350 CFU of either LVS, LVSΔ*fptA*, LVSΔ*fptF*, or PBS.

## Data Availability

All data is contained within the article or Appendix A.

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
