# Peer review of "Deletion Mutants of Francisella Phagosomal Transporters FptA and FptF Are Highly Attenuated for Virulence and Are Protective Against Lethal Intranasal Francisella LVS Challenge in a Murine Model of Respiratory Tularemia"

_pathogens, 2021, doi:10.3390/pathogens10070799_

Round 1

Reviewer 1 Report

In the work presented here, the authors seek to assess the vaccine efficacy of mutants of a putative phagosomal transporter family (Fpt) in Francisella tularensis species. The authors have previously evaluated mutants in fptB, E and G in the live vaccine strain (LVS) background using a Balb/c mouse model; these mutants had intracellular replication defects, were attenuated in the mouse model and provided protection against challenge with LVS. In the current manuscript, they evaluate mutants in additional genes fptA and fptF using a respiratory model of infection in a C57BL/6J background. The authors show that these mutants are attenuated but confer protection to lethal challenge with LVS. They further describe some of the immune parameters observed with the different mutants including induced cytokines and immune cell population in infected lungs.

Overall, the results are presented in a clear and coherent manner and support the conclusions. The manuscript may however be improved by clarifying and expanding on specific aspects as listed below.

  1. Lines 99-100 and 325-327. It would be helpful if the authors could provide some background regarding the FptA and FptF transporters (that were identified in a previous study). What characterization of the mutants was done with regard to growth in vitro? What are their possible functions, even if speculatively based on closest homologs in other species? How far does the in vitro phenotype reflect the vaccine efficacy phenotype seen?  Is there a sense of which Fpt mutants  have better efficacy overall? 
  2. Lines 239-240. The authors state that “Levels of IL-4, IL-5, and IL-6 were not significantly elevated in 239 any infection group versus mock infection (Supplementary Figure 1).” However, to this reviewer, it seemed strange that IL-6 levels were not significant, given that the mock infection levels were all zero and those in the infected groups, the levels were upwards of 1000 pg/mL.
  3. Lines 313-316. Several other types of mutants in metabolism have demonstrated protection to challenge with Type A strains in mice and the authors need to include references to them. Of note, one of the first mutants characterized in this class was the FupA mutant from Sjostedt lab. Additional mutants in iron transport and the ggt gene involved in cysteine metabolism have also demonstrated efficacy in mouse models.
  4. Figure 2 requires a key to the symbols for different strains.

Reviewer 2 Report

The authors of the current manuscript demonstrate that deletion of the ftpA and ftpE genes from F. tularensis LVS results in attenuated mutants with reduced virulence compared to the wild-type strain. Moreover, the mutants administered IN to C57BL6 mice protect them from a subsequent IN challenge with an otherwise lethal challenge of LVS. The authors then proceed to detail differences in in vivo growth, histopathology and immune responses elicited by LVS compared to the mutants and show significant differences in these parameters. This is a comprehensive and well-written manuscript.

Unfortunately, it suffers from one fatal flaw. Namely, it uses LVS as the challenge organism. This is a highly attenuated mutant of a type B strain of F. tularensis, and has been shown by multiple laboratories to be a poor surrogate for fully virulent type A strains in vaccination studies. Indeed, they admit to experiencing this problem themselves (line317-327). This is a pity since the senior author has made important contributions to tularemia vaccine development in the past using mutant strains of the organism that are exponentially more attenuated than both LVS ΔftpA and ΔftpE, yet capable of protecting mice from respiratory challenge with the highly virulent type A SCHU S4 strain. Therefore, the relatively high IN LD50 of the LVS mutants described herein render them far too virulent to be considered as vaccines for respiratory administration. Indeed, given that LVS is only approved for human use when administered by scarification, it would have been illuminating to know the ID LD50 of the LVS mutants and the protection that this route of administration afforded against respiratory challenge with LVS and type A F. tularensis strains. Given that the authors laboratory routinely uses the SCHU S4 strain in its prior studies, it is puzzling why they have chosen to use LVS as the challenge organism in the current study. The explanation in lines 75-80 simply do not reflect the reality of the massively different disease caused by virulent type A strains versus LVS. For instance, the former cause massive bacteremia and the latter does not. Also, antibodies alone can protect against LVS whereas a full-fledged CMI response is required to combat challenge with virulent type A strains.

If the authors do not want to undertake the relatively facile steps suggested above, then they should rewrite the article without the vaccine angle that is currently the key focus. If they choose this option then they ought to add a proviso in the discussion to the effect that attenuation in LVS by deletion of these genes does not guarantee a similar affect if deleted from virulent type A or B F. tularensis.

Another concern is the relatively high IN LD50 (<60 CFU) reported for the LVS strain obtained from the Elkins laboratory. Of the hundreds of papers reporting the IN or aerosol LD50 of LVS obtained from the Elkins laboratory including the Elkins group itself, and other sources of LVS, the consensus is an IN LD50 for BL6 mice somewhere in the region of 500 -5000 CFU. Moreover, in the present study, the LVS strain is grown using MHB which has been shown to reduce its virulence. Therefore, an explanation for the increased virulence of the LVS used in this study is required.

Reviewer 3 Report

The manuscript entitled “Deletion Mutants of Francisella Phagosomal Transporters FptA and FptF are Highly Attenuated for Virulence and are Protective Against Lethal Francisella Challenge in a Murine Model of Respiratory Tularemia” is a well written and very comprehensive efficacy study of deletion mutants of Francisella phagosomal transporters as a novel vaccine candidate against Ft. Study details show that FptA and FptF are highly attenuated and when given as a vaccine candidate, they provide protection from lethal intranasal challenge of Ft. in C57BL/6J mouse model. Below are some minor questions and suggestions.

1) Although there is a misconception of intranasal challenge considered as a respiratory route, it is not a true respiratory route. When animals are challenged intranasally, some of the challenge amount goes to their stomach. Thus, I recommend changing the title to

“Deletion Mutants of Francisella Phagosomal Transporters FptA and FptF are Highly Attenuated for Virulence and are Protective Against Lethal Intranasal Francisella Challenge in a Murine Model of Tularemia.

2) Table 1: How are vaccination doses chosen for each group? Why are they different?

3) Table 1: Why did groups inoculated with 1,120 and 1,680 FptA had 11 mice in them instead of 12 mice? There is an explanation for 670 FptF but not for the FPTA. Please add a footnote with explanation.

4) Results 2.1-Lines 129-133: Although mice were observed for clinical signs of the disease, there is no mention about any other clinical signs other than weight loss. Was there any other clinical signs such as reduced activity, ruffled fur etc?

5) Line 159: why was 350 CFU chosen instead of 500CFU?

6) Figure 2 is very difficult to read. How many mice are in each group at the beginning and how many mice had bacterial count in each time point for each groups? Very difficult to count overlapping shapes. In addition, the figure is missing legends. Panel (B) sign is overlapping with the x-axis title. Table with numerical info for each group might work out better.

7) Figure 3: Rulers cannot be seen easily. Need better resolution pictures and more contracts with rulers and the picture. Maybe a darker color ruler will work out better.

8) Line 267: Please insert (Figure 5A) after lungs, before period.

 9) Discussion: Lines 317-327 is a repeat of introduction. No need for this paragraph in discussion. Please delete.

10) Material and methods: Line 502: groups of 4-6 mice per dosage is not correct according to table 1.

11) Please make sure all the figure labeling are the same for the group names. Triangle indicating deletion in-group names is filled in Fig 5 and 6 while it is not filled in other figures. All figures should have unfilled version.

12) Authors did an extensive literature review, which is wonderful! However, I suggest keeping the references from last 20-30 years not 100 years.

Round 2

Reviewer 2 Report

The authors continue to maintain that respiratory challenge with LVS is a fair surrogate for respiratory challenge with virulent type A Francisella, and provide an extensive, but selective, literature review in support of this position. It would be churlish of me to respond with an equally selective literature review to the contrary. However, the abstract from another of the senior author’s papers (reference 62) states:

“In contrast, Schu S4 gua mutants were unable to replicate in murine macrophages and were attenuated in vivo, with an i.n. LD50 > 105 CFU in C57BL/6 mice. However, the gua mutants failed to protect mice against lethal challenge with WT Schu S4, despite demonstrating partial protection in rabbits in a previous study. These results contrast with the highly protectivecapacity of LVS gua mutants against a lethal LVS challenge in mice, and underscore differences between these strains and the animal models in which they are evaluated, and therefore have important implications for vaccine development”

Granted there are superficial similarities between murine tularemia caused by inhalation of LVS and type A F. tularensis in that both cause local infection and cellular inflammation and disseminate to the liver and spleen. However, similar results could be obtained with completely unrelated pathogens. Looking beyond the obvious similarities, the fact of the matter is that at the molecular level there are major differences between the host response to LVS and type A F. tularensis in mouse models and in vitro models using human tissues. These have been extensively documented by the Bosio group. Ironically, one key difference is that in mice, inhaled type A bacteria actively suppress pro-inflammatory responses in the lungs, whereas the authors ascribe the ability of LVS to cause more extensive pulmonary inflammation compared to the mutant strains as the cause of its enhanced virulence. Given this, and the fact that many live vaccines that protect against respiratory LVS challenge signally fail to do so against type A F. tularensis, the former seems to be a less than ideal screen for the latter, and a waste of mice. Be that as it may, I don’t want to start a to and fro battle of wits with the authors. Therefore, I believe that it should be considered for publication almost as is and allow the general readership to judge its merits. However, the authors need to change the title to read ”….intranasal Francisella LVS….”
